# The Effects of Different Kinds of Smooth Pursuit Exercises on Center of Pressure and Muscle Activities during One Leg Standing

**DOI:** 10.3390/healthcare10122498

**Published:** 2022-12-09

**Authors:** Zhe Cui, Ying-Ying Tang, Myoung-Kwon Kim

**Affiliations:** 1Department of Rehabilitation Sciences, Graduate School, Daegu University, Jillyang, Gyeongsan 712-714, Republic of Korea; 2Department of Physical Therapy, College of Rehabilitation Sciences, Daegu University, Jillyang, Gyeongsan 712-714, Republic of Korea

**Keywords:** smooth-pursuit eye movement, gaze fixation, center of pressure

## Abstract

This study examined the effects of gaze fixation and different kinds of smooth-pursuit eye movements on the trunk and lower extremity muscle activities and center of pressure. Methods: Twenty-four subjects were selected for the study. The activity of trunk and lower limb muscles (tibialis anterior, lateral gastrocnemius, medial gastrocnemius, vastus midialis obliques, vastus lateralis, biceps femoris, rectus abdominis, and erector spinae) and the COP (center of pressure) (surface area ellipse, length, and average speed) were measured to observe the effects of gaze fixation and different kinds of smooth-pursuit eye movements on the center of pressure and muscle activities during one leg standing. Before the experiment, a Gaze point GP3 HD Eye Tracker (Gazept, Vancouver, BC, Canada) was used to train eye movement so that the subjects would be familiar with smooth eye movement. Repeated each exercise 3 times at random. In order to avoid the sequence deviation caused by fatigue, the movement sequence is randomly selected. Result: The center of pressure and muscle activities were increased significantly when the smooth-pursuit eye movement with one leg standing compared with gaze fixation with one leg standing. In smooth-pursuit eye movements, the changes in the center of pressure and muscle activities were increased significantly with eye and head movement. When the head and eyes moved in opposite directions, the center of pressure and muscle activities were increased more than with any other exercises. Conclusion: Smooth-pursuit eye movement with one leg movement affects balance. In particular, in the smooth-pursuit eye movement with one leg standing, there were higher requirements for balance when the eyes and head move in the opposite direction. Therefore, this movement can be recommended to people who need to enhance their balance ability.

## 1. Introduction

Postural control is fundamental for performing numerous daily activities. The main behavioral goals involved in controlling posture are postural balance and postural orientation [1]. Balance skills play an axial and vital role in enjoying good physical health. Balance is a very complex function involving the musculoskeletal and nervous systems. The following functional factors play a vital role: the action of the central nervous system in response to stimulation from the sense of sight, sense of hearing, vestibular sense, proprioception sense, perceptivity of visual space, the myotonus responding quickly and accurately to environmental changes, muscular strength, endurance, and joint flexibility. Abnormalities in any of these factors may result in the loss of balance ability [2].

Balance is divided into static and dynamic balance. Static balance is the ability to maintain a base of support with minimal movement. Dynamic balance may be considered the ability to perform a task while maintaining or regaining a stable position [3]. Maintaining adequate static and dynamic balance is essential for body function and well-being across different ages. Adopting proper balance strategies to maintain equilibrium and have good postural control depends on task demand on balance performance [4]. Static balance skills, such as standing on one leg, are necessary for the activity of daily living. For example, during regular walking or walking up and down stairs, subjects need to shift their body weight on one leg and maintain balance while swinging the other leg. Standing on one leg is a simple test to assess static balance across different ages [4].

Vision has a primary role in governing spatial orientation and balance. The visual function is associated with the ability to maintain postural balance and perform daily activities [5]. Vision is an important sensory cue to familiarize ourselves with the external environment. A prerequisite is voluntary or involuntary eye movements, which are necessary to process information, such as recognition, localization, and proprioception. Vision also helps stabilize an upright posture by enabling the detection of self-motion relative to structures in the visual field [6]. Vision combined with eye movements transfers complex sensory information from the retina and extraretinal structures to the central nervous system. Eye movement will disturb the balance according to the movement [7].

Humans have three visual motion choices: visual fixation, saccades and smooth-pursuit eye movements (SPEM). Among them, SPEMs track small moving visual objects and depend on an intact fovea [8]. Eye movements can affect balance and various functions [7]. On the other hand, the previous studies considered eyeball exercise as a supportive method for improving the vestibular function and balance ability, rather than as a major method. In addition, previous studies focused on the central nervous system rather than the direct application of eyeball exercise [9]. Eye movement that affects posture control remain a controversial issue, especially for smooth pursuit eye movements [10].

Therefore, this study examined the effects of different kinds of smooth-pursuit eye movements on the trunk and lower extremity muscle activities and center of pressure (surface area ellipse, length, and average speed). The measured muscles were the tibialis anterior (TA), rectus abdominis (RA), medial gastrocnemius (MG), lateral gastrocnemius (LG), erector spinae (ES), vastus midialis obliques (VMO), vastus lateralis (VL), and biceps femoris (BF).

## 2. Methods

### 2.1. Experimental Subjects

The sample size was estimated through an a priori power analysis carried out with the G* Power program (G power program Version 3.1, Heinrich-Heine-University Düsseldorf, Germany) assuming an univariate approach for within effects. For the procedure the following parameters were taken into account: effect size = 0.8 (based on data from a pilot study), alpha = 0.05, power = 0.80. To account for a potential dropout rate of 20% the estimated sample size (n. 20) has been implemented at 24. Before participating, all subjects read and signed university-approved human subject consent forms. The Institutional Review Board of Daegu University approved the study (IRB No. 1040621-202209-HR-077). The inclusion criteria were as follows: (1) Those who did not participate in another balance training program for more than three times a week in the last six months; (2) those who do not have any neurologic problems, including damage to the central nervous system or vestibular system; (3) those with no orthopedic problems including fractures or severe osteoarthritis; (4) those with no visual perceptual disorders [9]. The exclusion criteria were as follows: (1) subjects with a history of visuomotor impairment (or eye movement dysfunctions), anisometropia, amblyopia, nystagmus, or other abnormal binocular functions; (2) subjects with somatosensory defects that affect the balance ability, and (3) subjects with a degenerative disease and musculoskeletal system impairment affecting the standing balance [2,7].

### 2.2. Experimental Procedure

#### 2.2.1. Experimental Design

Twenty-four subjects were selected for the study. The activity of trunk and lower limb muscles (TA, LG, MG, VMO, VL, BF, RA, and ES) and the COP (surface area ellipse, length, and average speed) were measured to observe the effects of gaze fixation and different kinds of smooth-pursuit eye movements on balance and muscle activities during one leg standing. Before the experiment, Gazepoint GP3 HD Eye Tracker (Gazept, Vancouver, BC, Canada) was used to train eye movement so that the subjects were familiar with smooth eye movement. The overall eyeball exercise program was reconstructed based on the eyeball exercise suggested in a previous study [11]. Four different exercises were performed: (1) gaze fixation (GF) with one leg standing—raise their non-dominant leg, put hands on both sides of legs and keep eyes on the front; (2) smooth-pursuit eye movement exercise (SPEM) 1—when standing on one leg, move the target horizontally and track it with the eyes while keeping the head still; (3) smooth-pursuit eye movement 2—when standing on one leg, move the head and target in the same directions horizontally while tracking the target with the eyes; (4) smooth-pursuit eye movement 3—when standing on one leg, move the head and target in opposite directions horizontally while tracking the target with the eyes. Repeated each exercise 3 times at random. In order to avoid the sequence deviation caused by fatigue, the movement sequence is randomly selected (Figure 1) [12].

#### 2.2.2. Smooth-Pursuit Eye Movements

The subjects stood 70 cm in front of the monitor and stared at the screen. When the video is run, dots will move after three seconds. The dots moved gently from left to right and from right to left, and the subjects stared at the moving dots with their heads fixed or unfixed. During head rotation, the head rotation angle was limited to 30° (Figure 2).

#### 2.2.3. Gazepoint GP3 HD Eye Tracker

In this study, the Gazepoint GP3 HD Eye Tracker (Gazept, Vancouver, BC, Canada) was used to ensure that the subject’s eyes could target horizontally and track objects during the experiment and participate correctly in the experiment. Eye tracker can collect binocular data of subjects at sampling rates of 60 Hz and 150 Hz. It has two infrared light-emitting diodes (LEDs) that provide light that would reflect on the cornea of the eye to detect eye movement. The eye tracker is able to work at an optimal distance of 65 cm away from the eye (Figure 3).

#### 2.2.4. BPMpro Technology

BPMpro Technology (Brown Spring, Pelican Products, South Deerfield, MA, USA) is a tool to visualize the movement of joints, and it is printed out on a PC screen in 3D animation. The equipment is placed in the middle of the forehead and used to limit head rotation to 30° in the experiment.

### 2.3. Measurements

#### 2.3.1. Bio-Rescue (RM Ingenerie, France)

A balance assessment device (BioRescue, RM Ingenieria, France), consisting of a force plate equipped with sensors for static balance evaluation in standing position. The subject’s feet are approximately 30 feet apart on a force plate, and images on a display mounted in front of the subject provide a description. The measurements were taken after the demonstration. The surface area ellipse (mm^2^), length (cm), and average speed (cm/s) of body sway were measured during the different eye movements. The average of the three repeated measurements was used to calculate the surface area ellipse [13].

#### 2.3.2. Surface electromyography (EMG)

(1) Signal Collection and Analysis

A 16-channel radio-surface electrocardiogram (TeleMyoDTS, NoraxonIns, AZ, USA) was performed to collect the myocardiogram data and process signals.

The collected electrodynamic analog signals are sent to the Telemyo system DTS and converted into digital signals. Other signals are filtered and processed using myre search XP 1.08 software (NoraxonIns, AZ, USA). The body surface ECG signal is filtered, and then other signals are processed by personal computer. The sample rate is set to 1500 Hz to measure activity. The frequency bandwidth is filtered with a 40~450 Hz band-pass filter and a 60 Hz notch filter to quantify all collected myocardial conduction signals [12]. The signals collected for each muscle were processed as a valid mean value (RMS) and as a percentage (%) of the maximum normal length contraction.

The collected electro-dynamic analog signals are sent to Telemyo system DTS and converted into digital signals. Other signals are filtered and processed using myre Search XP 1.08 software (NoraxonIns, AZ, USA). The surface ECG signals are filtered and then processed with a personal computer. The sampling rate was set to 1500 Hz to measure activity. Bandpass filters of 40~450 Hz and notch filters of 60 Hz were used to filter the frequency bandwidth to quantify all the collected cardiac conduction signals [12]. The signals collected for each muscle were processed as an effective mean value (RMS) and as a percentage (%) of the maximum value of normal length contraction.

Telemyo system DTS converted the collected electrodynamic analog signals into digital signals, and then filtered and processed other signals using Myoresearch XP 1.08 software (NoraxonIns, AZ, USA). Surface electrocardiogram signals were filtered and other signals were processed with a personal computer. To measure muscle activity the sampling rate was set at 1500 Hz. Bandpass filters of 40~450 Hz and notch filters of 60 Hz were used to filter the frequency bandwidth to quantify all the collected cardiac conduction signals [12]. The signals collected for each muscle were processed as an effective mean value (RMS) and as a percentage (%) of the maximum value of normal length contraction.

(2) Normalization

The measurement was carried out using the method reported by Choi and Kang [14]. Electromyography signals were collected for 5 s for each action. The first and last second of data is deleted; It only took three seconds to analyze. Electromyography was measured three times and averaged. Allowed a one-minute break between each measurement. Experimental data were expressed as percentage of MVIC calculated root mean square (% MVIC). The maximum voluntary isometric contraction (MVIC) activity of the measured muscle was measured to normalize Electromyography activity. The method of measurement is described elsewhere [15]. For the TA, the subjects were seated and maximally dorsiflexed and inverted their foot against a rigid strap. For the LG and MG, the subjects in the prone position, plantar flexed the foot forcefully against a resistance. For the VMO and VL, the hip joint and knee joint flexed to 90 degrees in a sitting position and were instructed to extend the knee forcefully to resist resistance. For the BF, the hip joint and knee joint flexed to 90 degrees in a sitting position and were instructed to flex knee forcefully against a resistance. For the RA, the subjects in the supine posture performed a curl-up forcefully against a resistance. For the ES, the subject in the prone position lifted the upper part of the body forcefully against a resistance.

(3) The placement of the surface electrode

The surface electrode was placed on the dominant leg of the subject. The dominant leg depends on which leg the participant is used to kicking the ball [16].The skin needs to be shaved and wiped with disposable medical alcohol before sticking the electrode to ensure that the skin is clean and smooth. Placed the surface electrode pair at an interelectrode distance of 2 cm. For the TA, electrodes were placed about one-third of the way down the line between the tip of the fibula and the tip of the medial malleolus. For the VL, the electrode is placed about 3–5 cm above the patella, and is inclined at the outside of the midline. For the BF, two thirds of the distance between the trochanter and the back of the knee. For the VMO, electrodes were placed at an oblique angle (55°), 2 cm medially from the superior rim of the patella. The palpate for the muscle during the knee was extended. The electrodes were placed on the distal third of the vastus medialis oblique. For the LG and MG, the electrodes were placed at the distal end of the knee joint, 2 cm inside or outside the midline. For the ES, approximately 2 cm from the spine over the muscle mass. The iliac crest can be used to determine the L-3 vertebra. For the RA, the electrodes were placed 3 cm apart, parallel to the rectus muscle fibers, so they were located approximately 2 cm lateral to and across from the umbilicus over the muscle belly [17].

## 3. Statistical Analysis

SPSS 20.0 software (SPSS Inc., Chicago, IL, USA) was used for all statistical analyses. One-way repeated ANOVA was applied to determine the significant difference in the muscle activity and center of pressure. The Bonferroni correction method was used to control for multiple comparisons. The level of significance was 0.05

## 4. Results

The general characteristics of the subjects are shown in Table 1.

Significant differences were observed in the surface area ellipse, average speed, and length (*p* < 0.05). Three different kinds of smooth-pursuit eye movements with one-leg standing exercises were significantly higher than gaze fixation with OLS. SPEM3 with the surface area ellipse of OLS, average speed, and length were higher than SPEM1 and SPEM2 with OLS (Table 2) (Figure 4).

Significant differences in the TA, LG, MG, and VMO muscle activities were observed among the exercises (*p* < 0.05). However, no significant difference was observed in the muscle activity of VL, BF, ES and RA among the four exercises (*p* > 0.05). In the TA and MG, the three different kinds smooth-pursuit eye movements with one leg standing exercises produced significantly higher EMG activation than gaze fixation with OLS. SPEM3 with the OLS muscle activities showed higher EMG activation than SPEM1 and SPEM2 with OLS. In LG, the three different kinds of smooth-pursuit eye movement with one-leg standing exercises produced significantly higher EMG activation than gaze fixation with OLS, but there was no significant difference between the three different smooth-pursuit eye movements. In the VMO, three different smooth-pursuit eye movements with one-leg standing exercises produced significantly higher EMG activation than gaze fixation with OLS. Furthermore, the SPEM3 with OLS muscle activities were higher than SPEM2 with OLS (Table 3) (Figure 5).

## 5. Discussion

Good balance is essential for daily life, which requires the complex integration of sensory information on the body position relative to the surroundings and the ability to generate appropriate motor responses to control body movement [18]. Vision is an important factor for balance control and mobility. Visual function is associated with the ability to maintain postural balance and perform daily activities [5]. This study examined the effects of gaze fixation and different kinds of smooth-pursuit eye movements on the trunk and lower extremity muscle activities and center of pressure (surface area ellipse, length, and average speed).

In this study, the center of pressure (surface area ellipse, length, and average speed) and muscle activities were increased significantly when smooth-pursuit eye movement with one-leg standing was performed compared to gaze fixation with one-leg standing. During smooth-pursuit eye movements, the postural sway and muscle activities were increased significantly with eye and head movement when the head and eyes moved in opposite directions. The center of pressure (surface area ellipse, length, and average speed) and muscle activities increased more than any other exercise.

Thomas et al. [6] studied the impact of eyeball exercises and gaze-fixing stability exercises on adults and the stability of posture. They reported a larger increase in postural sway after eye movements than after eye fixation. This shows that eyeball exercise and gaze fixing stability exercise influenced posture stability. During smooth pursuits, the image of the visual target may appear stable on the fovea [19], but the background visual field shifts on the retina opposite the target movement. This would generate similar retinal flow patterns to an oscillating background visual field, which may lead to more challenging conditions for estimating the body position [20]. This is consistent with the experimental results. The center of pressure (surface area ellipse, length, and average speed) increased significantly when the smooth-pursuit eye movement with one leg standing compared to gaze fixation with one leg standing.

There is a close relationship between how visual and vestibular information about the head position is used for postural control [21]. Previous studies have reported that active head movements decrease the visual acuity, postural control, and postural stability in patients with a vestibular dysfunction and healthy subjects [22,23]. In addition, slow gaze movements, performed by the eyes alone or combined eye-head movements, can induce significant body sway [24]. Some authors have reported that head movements stimulate the vestibulo-ocular reflex, producing eye movements at the same speed but in opposite directions to maintain vision [25]. Rapid head movements can lead to retinal slippage, and vision loss occurs when retinal slippage exceeds 2–4/s [11]. This study showed the same results. Compared to smooth-pursuit eye movement without head movements, smooth-pursuit eye movement with head movements showed higher postural instability and greater body swaying power.

Nakashima and Shioiri [26] reported that visual performance was higher when the head and eyes were oriented in the same direction versus different directions. Some studies have reported that hand position perception or the accuracy of depth estimation may be reduced when the eye and head directions are different. They believed that visual sensory inputs are encoded in different reference frames (eye centered, head centered). If the head and eyes point in different directions [27,28,29], the spatial perception accuracy will decline. The misplacement of the reference frame between the eyes and the head will make the attention point to the direction of the head or the location of the gaze. When the head points to the visual stimulus, the visual performance is higher than that of non [27,28]. When the attention direction is different from the eye direction, it can be divided according to the head direction, and the weight of the eye direction may be greater. When the eyes and head are in the same direction, the processing efficiency can be improved. This may be because observers tend to turn their heads around to visual stimuli in tasks, they think are problematic. To accomplish a difficult task, they may want to focus on resources by orienting their eyes and head [30]. These results showed that differences in head and eye orientation interfere with attentional processing in visual search. The accuracy of concentration may be reduced when the head and eyes are in different directions. Therefore, in this experiment, the COP increased the most, and body imbalance was highest when the direction of eye movement was opposite to the direction of head movement during smooth-pursuit eye movement.

This adjustment occurs through movements of the ankles, knees, and hips. It may be disturbed when the center of gravity and base of support is disrupted or when corrective movements are not executed in a smooth and coordinated fashion [31]. Information from the plantar of the feet and the leg muscles contributes significantly to upright standing [5]. The ankle joint has the function of balance control and can respond to the reaction force of the ground; It contracts the tibialis anterior, soleus and gastrocnemius muscles [32]. The plantar dorsal flexors and flexors play a central role in maintaining the balance between single limb support and dual limb support. Soleus and gastrocnemius muscles play an important role in posture correction [33]. Muscle strength and muscle activation are balanced to keep the center of gravity on the basis of support [34]. Co-activation of the quadriceps and hamstrings increases joint stiffness and maintains stability [31]. This is consistent with the results. During the experiment, smooth-pursuit eye movement with one leg standing increased the body sway more than gaze fixation with one leg standing, and the body instability increased. The ankle and thigh muscles work together to increase the joint stability and balance the body. Therefore, the muscle activities of smooth-pursuit eye movement were significantly higher than that of gaze fixation. Active head movement reduced the subject’s posture control and stability in smooth-pursuit eye movement. When the head and eyes are facing the same direction, the visual expressiveness will be higher than that in different directions. That is, the swing and imbalance of the body are the highest when the head and eyes are in different directions. Therefore, ankle and thigh muscles are maximally activated to maintain a stable posture while standing on one leg.

This study had some limitations. First, the study population included only students, not other professional groups. The small size may have influenced certain variables and influenced the results. Second, this study is a cross-over design, it is difficult to evaluate the long-term effects of smooth-pursuit eye movement on muscle activities and balance. Third, the surface EMG for data collection was used. Crosstalk from the surrounding muscles may produce erroneous signals when using surface EMG electrodes. Future research should improve this limitation by increasing the number of subjects and selecting patients who require balance ability rather than young and healthy people. In addition, a longitudinal study, rather than a cross-sectional study, is needed from multiple perspectives.

## 6. Conclusions

This study examined the effects of gaze fixation and different kinds of smooth-pursuit eye movements on the trunk and lower extremity muscle activities and center of pressure. The center of pressure and muscle activities increased significantly when the smooth-pursuit eye movement with one leg standing compared to gaze fixation with one leg standing. During smooth-pursuit eye movements, the center of pressure and muscle activities were increased significantly with eye and head movement when the head and eyes move in opposite directions, the center of pressure and muscle activities were increased more than any other exercise. As a result, smooth-pursuit eye movement with one leg movement affected balance. In particular, in the smooth-pursuit eye movement with one leg standing, they had higher requirements for balance when the eyes and head move in the opposite direction. Therefore, this movement can be recommended for subjects who need to enhance their balance ability.

## Figures and Tables

**Figure 1 healthcare-10-02498-f001:**
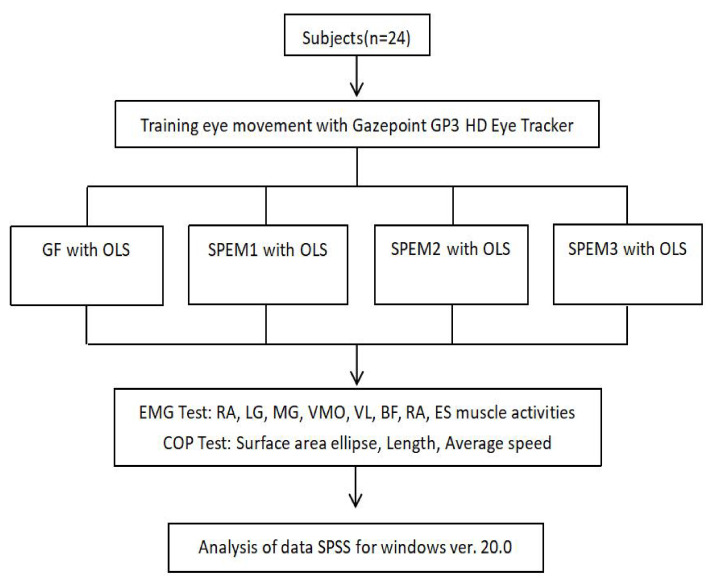
Study flow chart. GF: gaze fixation. OLS: one leg standing. SPEM: smooth-pursuit eye movement exercise.

**Figure 2 healthcare-10-02498-f002:**
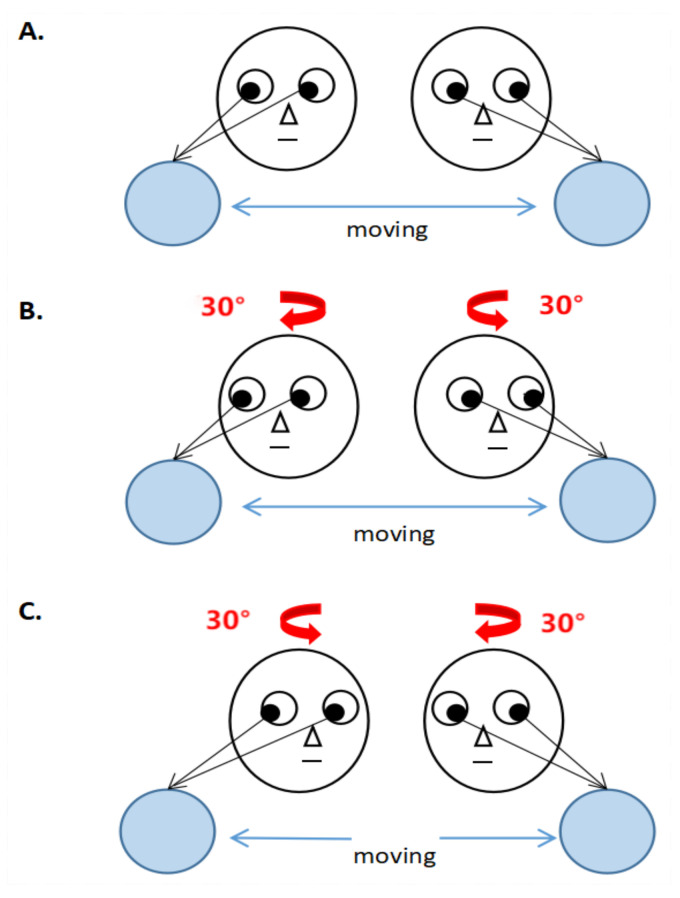
Smooth-pursuit eye movement exercises ((**A**): Smooth-pursuit eye movement exercise 1—when standing on one leg, move the target horizontally and track it with the eyes while keeping the head still; (**B**): Smooth-pursuit eye movement 2—when standing on one leg, move the head and target in the same directions horizontally while tracking the target with the eyes; (**C**): Smooth-pursuit eye movement 3—when standing on one leg, move the head and target in the opposite directions horizontally while tracking the target with the eyes.).

**Figure 3 healthcare-10-02498-f003:**
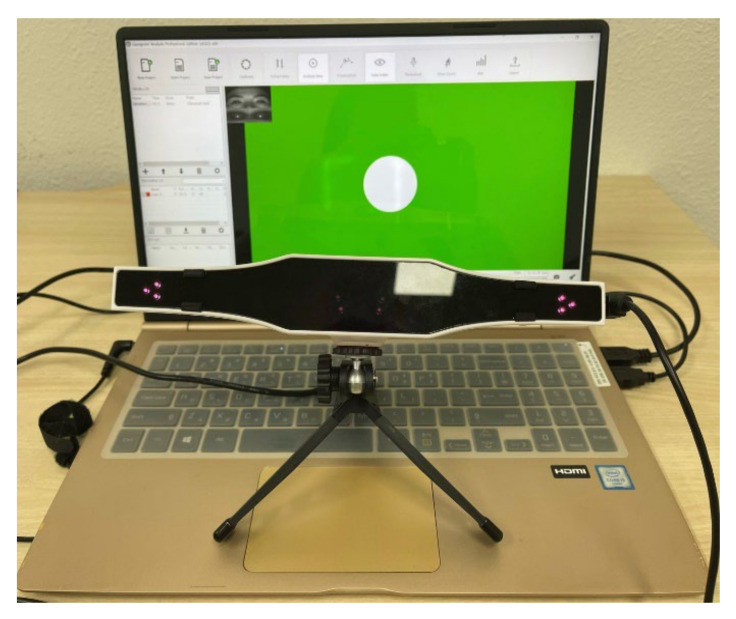
Gazepoint GP3 HD Eye Tracker.

**Figure 4 healthcare-10-02498-f004:**
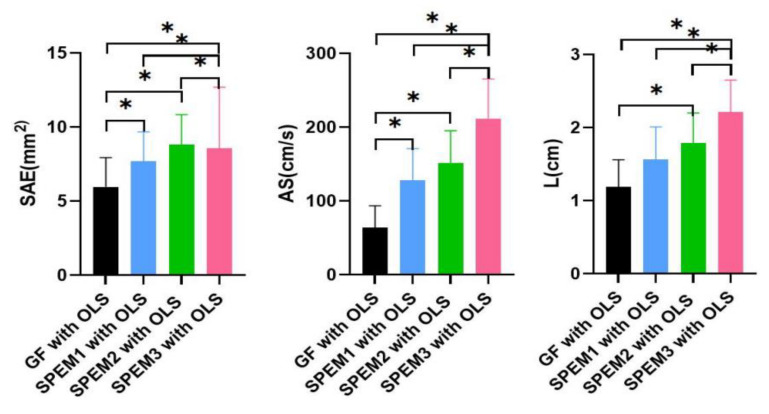
Comparison for four exercises on the center of pressure (surface area ellipse, length, and average speed). * Statistical significance *p* < 0.05.

**Figure 5 healthcare-10-02498-f005:**
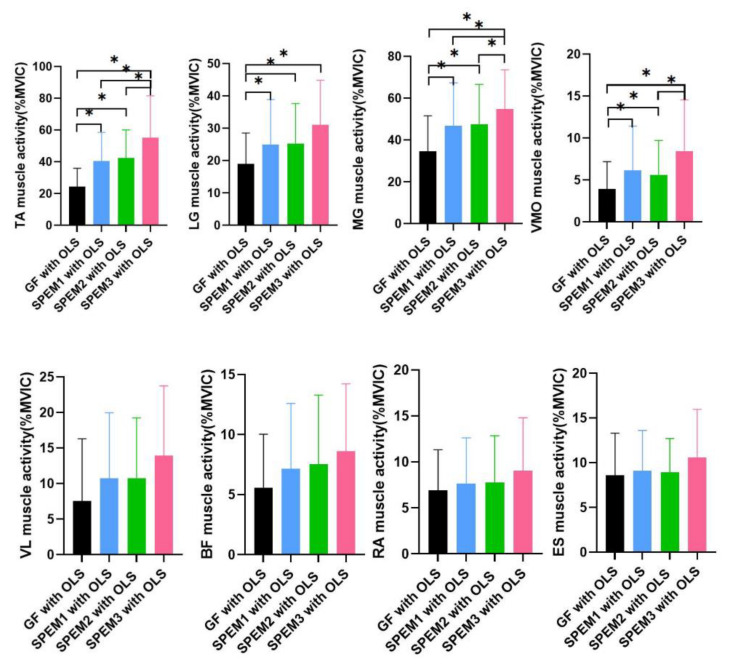
Comparison of four exercises on the trunk and lower extremity muscle activity. * Statistical significance *p* < 0.05.

**Table 1 healthcare-10-02498-t001:** General characteristics of the subjects.

Variable	Subjects
Age (year)	26.43 (3.89) ^1^
Height (cm)	169.39 (7.25)
Weight (kg)	67.57 (14.60)
BMI ^2^ (score)	23.40 (3.92)

^1^ Mean (±standard deviation). ^2^ Body Mass Index.

**Table 2 healthcare-10-02498-t002:** Comparison of the four exercises on the center of pressure (surface area ellipse, length, and average speed).

	GF ^4^ with OLS ^5^	SPEM1 ^6^ with OLS	SPEM2 with OLS	SPEM3 with OLS	F	*p*
SAE ^1^	5.97 (1.97) ^7^	7.68 (2.00)	8.80 (2.04)	8.58 (4.12)	11.221	0.000 *
AS ^2^	64.24 (29.30)	128.07 (43.16)	151.67 (43.62)	211.98 (53.44)	45.894	0.000 *
L ^3^	1.19 (0.37)	1.56 (0.45)	1.79 (0.41)	2.21 (0.44)	23.894	0.000 *

^1^ Surface area ellipse. ^2^ Average speed. ^3^ Length. ^4^ Gaze fixation. ^5^ One leg standing. ^6^ Smooth-pursuit eye movement. ^7^ Mean (±standard deviation). * Statistical significance *p* < 0.05.

**Table 3 healthcare-10-02498-t003:** Comparison of four exercises on the trunk and lower extremity muscle activity.

Muscles	GF ^9^ with OLS ^10^	SPEM1 ^11^ with OLS	SPEM2 with OLS	SPEM3 with OLS	F	*p*
TA ^1^	24.44 (11.49) ^12^	40.46 (18.04)	42.42 (17.70)	55.33 (26.31)	9.652	0.000 *
LG ^2^	19.04 (9.52)	24.99 (13.90)	25.3 (12.40)	31.07 (13.82)	3.377	0.022 *
MG ^3^	34.56 (17.10)	46.75 (20.64)	47.53 (19.08)	54.76 (18.90)	4.291	0.007 *
VMO ^4^	3.93 (3.26)	6.17 (5.25)	5.62 (4.08)	8.46 (6.09)	3.343	0.023 *
VL ^5^	7.56 (8.75)	10.71 (9.26)	10.78 (8.48)	13.92 (9.84)	1.793	0.155
BF ^6^	5.60 (4.43)	7.14 (5.45)	7.53 (5.76)	8.61 (5.62)	1.198	0.316
RA ^7^	6.90 (4.43)	7.66 (4.97)	7.79 (5.06)	9.04 (5.77)	0.673	0.571
ES ^8^	8.59 (4.72)	9.07 (4.53)	8.92 (3.79)	10.58 (5.40)	0.795	0.500

^1^ Tibialis anterior. ^2^ Lateral gastrocnemius. ^3^ Medial gastrocnemius. ^4^ Vastus midialis obliques. ^5^ Vastus lateralis. ^6^ Biceps femoris. ^7^ Rectus abdominis. ^8^ Erector spinae. ^9^ Gaze fixation. ^10^ One leg standing. ^11^ Smooth-pursuit eye movement. ^12^ Mean (±standard deviation). * Statistical significance *p* < 0.05.

## Data Availability

All data related to this study are included in the article.

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
