# Peer review of "The Effects of Different Kinds of Smooth Pursuit Exercises on Center of Pressure and Muscle Activities during One Leg Standing"

_healthcare, 2022, doi:10.3390/healthcare10122498_

Round 1

Reviewer 1 Report

General comments

Considering how much the postural control is fundamental for performing numerous daily activities and that the balance is achieved and maintained by a complex set of sensorimotor control systems, the commitment of the authors in dealing with such a complex issue is to be appreciated.

Despite the study has some limitations, clearly indicated by the authors in the discussions, the results addressed are important and provide suggestions for subjects who need to enhance their balance ability.

Nevertheless, there are some points that need revision. (See specific Comments)

 Specific comments

Line 63           define SPEM (has not been done before)

Line 78-82      Sample size estimate

Please rewrite as follow

The sample size was estimated through an a-priori power analysis carried out with the G* Power program  (G power program Version 3.1, Heinrich-Heine-University Düsseldorf, Germany) assuming an univariate approach for within  effects. For the procedure the following parameters were taken  into account: effect size = ??? (based on data from a pilot study), alpha = 0.05, power = 0.80. To account for a potential dropout rate of 20% the estimated sample size (n. 20) has been implemented at 24.

Line 101              OLS has not been defined;  also how the dominant leg was determined?

Line 116               Figure 1. Study flow chart. Please define OLS and SPEM

Line 221 – 225  Statistical analysis

Please rewrite specifying the procedure in more detail.

Why did the authors decide for multiple comparisons (post hoc) to use the LSD test that did not make any correction to the error rate for multiple comparisons?

Author Response

We revised the paper according to the reviewer's comments. 

Thank you. 

Reviewer 2 Report

This study investigating the the effects of gaze fixation and different kinds of smooth pursuit eye movements on the trunk and lower extremity muscle activities and center of pressure. They used Gazepoint GP3 eye tracker for tacking the eye movements. This is an interesting study but It would be better to give more details about the usage of eye tracker. It was mentioned as "In this study, the Gazepoint GP3 HD Eye Tracker (Gazept, Vancouver, Canada) was used to ensure that the subject's eyes could target horizontally and track objects during the experiment and participate correctly in the experiment." Authors should mention about the details of the eye tracker with its Hz information etc. also it is better to mention about a calibration process before starting to the experiment. For better understanding the experimantal procedure it would be good to provide photo from the experiment scene.

Author Response

(The authors gave the same response as above.)
